# Recent Progresses in the Catalytic Stereoselective Dearomatization of Pyridines

**DOI:** 10.3390/molecules28176186

**Published:** 2023-08-22

**Authors:** Lucrezia Margherita Comparini, Mauro Pineschi

**Affiliations:** Department of Pharmacy, University of Pisa, Via Bonanno 33, 56126 Pisa, Italy; lucrezia.comparini@phd.unipi.it

**Keywords:** dearomatization, pyridine, pyridinium salt, asymmetric catalysis, dihydropyridine, tetrahydropyridine, piperidine, 2-pyridone, 4-pyridone

## Abstract

1,2- and 1,4-dihydropyridines and *N*-substituted 2-pyridones are very important structural motifs due to their synthetic versatility and vast presence in a variety of alkaloids and bioactive molecules. In this article, we gather and summarize the catalytic and stereoselective synthesis of partially hydrogenated pyridines and pyridones via the dearomative reactions of pyridine derivatives up to mid-2023. The material is fundamentally organized according to the type of reactivity (electrophilic/nucleophilic) of the pyridine nucleus. The material is further sub-divided taking into account the nucleophilic species when dealing with electrophilic pyridines and considering the reactivity manifold of pyridine derivatives behaving as nucleophiles at the nitrogen site. The latter more recent approach allows for an unconventional entry to chiral *N*-substituted 2- and 4-pyridones in non-racemic form.

## 1. Introduction

The dearomatization of heteroaromatic compounds is a very powerful synthetic strategy that can also build up complex structures in an asymmetric fashion starting from relatively abundant substrates [1,2]. Nonaromatic six-membered azaheterocycles are privileged structures present in a variety of natural compounds, small molecule drugs, and agrochemicals [3]. In this context, the synthesis and applications of dihydropyridines continue to attract considerable interest *di per se* and also due to the presence of two double bonds that can be further elaborated in a regio- and stereoselective fashion to give highly substituted tetrahydropyridines and piperidines [4] or to impart molecular diversity in a DOS approach [5,6,7]. *N*-Substituted 2-pyridones are also important compounds found in many natural products and medicinally relevant molecules [8,9] that can be obtained in a stereoselective fashion by the use of dearomative asymmetric procedures. 

Aside from starting with acyclic precursors such as in Hantzsch-type synthesis [4], one of the most straightforward general methods to obtain hydropyridine derivatives in a stereoselective fashion relies on the dearomatization of pyridine derivatives. Most often the dearomatization process can be achieved by the use of pyridine as the electrophilic reaction partner in nucleophilic addition (Figure 1a). Nucleophilic additions to activated pyridines, generally known as a Reissert-type reaction, are a largely explored reactivity that have mainly focused on the dearomatization reactions of substituted *N*-acyl or *N*-alkylpyridinium ions [10]. Much less explored is a complementary strategy where the stereoselective dearomatization occurs by means of pyridine derivatives that behave as nucleophiles at the nitrogen site (Figure 1b). So far, the latter strategy has been mostly developed via 2-substituted pyridine derivatives leveraging inter- and intra-molecular allylic amination reactions or sigmatropic rearrangements to yield chiral 2-pyridones. 

As the former approach to dealing with nucleophilic addition to activated electrophilic pyridines has recently been reviewed up to mid-2018 [11], this review presents only the most recent literature about this topic. The asymmetric hydrogenation [12] or transfer hydrogenation [9] of substituted pyridines to give piperidines, and the use of pyridinium ylides to effect cyclization processes [11] are not covered in this review. 

The material presented herein is classified according to the type of pyridine reactivity (electrophilic/nucleophilic) and to the type of catalyst and/or nature of the reactions. 

## 2. Catalytic Stereoselective Dearomatization of Pyridine Scaffolds with Nucleophiles

### 2.1. Metal Catalysts

Nonaromatic nitrogen-containing heterocycles such as dihydropiridine (DHP) and piperidine structures are extremely common in approved drugs, agrochemicals, and alkaloid natural products [2]. For these reasons, organic chemists are interested in the development of new ways to obtain these ubiquitous scaffolds, especially in a stereoselective fashion and using milder reaction conditions then those used previously [10,13,14,15,16,17,18]. Various methodologies were developed for the asymmetric synthesis of these frameworks, but in the last few years catalytic asymmetric dearomatization has gained the most interest, probably due to the convenient use of commercially available and inexpensive starting materials such as pyridine and quinoline [1,10,19]. The challenge of asymmetric dearomatization is the low reactivity of these scaffolds, which usually need to be activated to be reactive towards the addition of nucleophiles. For this reason, the activation of pyridine as a pyridium salt and the use of a transition-metal catalyst is the most common strategy to achieve dearomatization, and several chiral ligands have been employed to obtain stereocontrol with a degree of success [20,21,22,23,24]. Moreover, in the last few years a great interest in the use of an organocatalytic approach with an activated pyridinium ion has risen as well [25,26,27,28,29]. In this review the latest (from the second half of 2018) reports of the nucleophilic asymmetric dearomatization of pyridine are presented, to provide an updated overview of the state of the art for the synthesis of enantioenriched DHPs and piperidines, as well as complex multicyclic and spiro-compounds.

#### 2.1.1. Grignard Reagents

In 2019, Yu and coworkers presented a new regioselective and diastereoselective access to tetrahydropyridines and tetrahydroquinolines through a one-pot double nucleophilic addition of Grignard reagents to pyridine (**1**) and quinoline derivatives (**4**) with organoborane catalysts [30]. This method allows for the obtaining of the corresponding tetrahydropyridines and tetrahydroquinolines with two carbon nucleophiles via direct C-C bond-forming dearomative reactions (Figure 1). The protocol requires the use of BF_3_-Et_2_O at 0 °C to obtain the corresponding boron trifluoro salt **2** and **5**, then the first nucleophile (Grignard reagent) is added at −50 °C to exclusively form the 1,4-derivative. The addition of MeOH allows for the protonation of the formed enamine, resulting in an electrophilic iminium ion. The addition of the second nucleophile and TFA allows the desired product to be obtained with complete regio- and stereocontrol; with good to excellent yields; and with several functional groups and substituents in positions 3, 5, and 7 as well. In these conditions, the authors found that only the *anti*-diastereoisomeric product **6a** was formed for quinoline derivatives, while at room temperature, the stereoselectivity was reversed, resulting in the *syn*-diastereoisomer **6b**. The stereoselectivity of the reaction is controlled by the second nucleophilic attack and is kinetically controlled at low temperatures, while at room temperature the mixture is in equilibrium and the thermodynamic product is obtained. The study was extended to pyridine derivatives **1** to obtain the desired product with complete regioselectivity and good yields. In most cases, with pyridine scaffolds a good diastereoselectivity was obtained and both at low and room temperature the main product of the reaction was the *anti*-diastereoisomer **3a**, probably due to the lower stability of the iminium ion that makes the reaction irreversible in both conditions. A broad range of Grignard reagents was screened for this reaction, all giving moderate to good yields and very good stereoselectivity. Various kinds of second nucleophiles were also tested to obtain a set of diversely substituted tetrahydropyridines and tetrahydroquinolines that are privileged structures in medicinal chemistry.

A highly enantioselective catalytic dearomatization of in situ-formed *N*-acylpyridinium salts with Grignard reagents and copper catalysts was presented in 2021 by Harutyunyan’s group [31]. The dearomatization of 4-methoxypyridine was carried out at −78 °C, where (*R*,*R*)-Ph-BPE **L1** (6 mol%), CuBr-Me_2_S (5 mol%), and Grignard reagent (2.0 equiv) were in toluene for 12 h (Figure 2). Several acylating agents were tolerated, with benzyl chloroformate generating the higher ee (99%) and higher yields. In this protocol, an increased temperature reduced the enantioselectivity, but a wide variety of Grignard reagents were tolerated, except for secondary Grignard reagents that gave a racemic mixture; 2- and 3-substituted 4-methoxypyridines were also used as substrates, using methyl chloroformate in CH_2_Cl_2_ to generate synthetically useful yields (51–66%) and high enantioselectivities (80–97%) of the corresponding dihydro-4-pyridones **9**. An appealing upside to the reaction is the possibility of maintaining good results (68% yield, 97% ee) even in a gram-scale reaction. Mechanistic studies revealed that the high enantioselectivity derives from the bidentate nature of the ligand and from the flexible linkage between the binding arms that allow the transfer of chiral information from the catalyst to the product. 

In a study very recently reported by the same group, a novel and enantioselective direct method for the C4-dearomatization of 2-methoxypyridine derivatives of type **10** was reported [32] (Figure 3). The aim of the work was the preparation of functionalized enantioenriched δ-lactams, highly valuable molecular frameworks in medicinal chemistry with well-established pharmacological properties [33,34,35]. This approach involved the synergistic use of copper(I) catalysis, Grignard reagents, and a strong Lewis acid (BF_3_·Et_2_O), which interacted with the nitrogen atom of 2-methoxypyridine to enhance its electrophilicity in situ, eliminating the need for the formation of a pyridinium ion to create chiral δ-lactam derivatives **11**, with excellent control over their regio- and stereochemistry (Figure 3). The optimized reaction conditions consisted CuBr·SMe_2_ (5.0 mol%), (*R*,*R*)-Ph-BPE **L_1_** (6.0 mol%), BF_3_·Et_2_O (1.2 equiv), and Grignard reagent (1.2 equiv) in CH_2_Cl_2_ at −78 °C for 16 h. The method demonstrated the compatibility of a wide range of Grignard reagents and substituents with the 2-methoxypyridine derivatives if not at the C4 or C2 position, delivering high yields and exceptional enantioselectivities of up to 99% ee, while maintaining high regiocontrol. For reactions involving PhMgBr, lower enantioselectivities could be overcome by increasing the catalyst loading. The researchers employed density functional theory (DFT) calculations and ^13^C kinetic isotope effect (KIE) studies to gain insights into the reaction mechanism. The DFT study confirmed that the Lewis acid, upon complexation with the pyridine substrate, played a crucial role in activating the system electronically, facilitating the addition of nucleophilic species through the chiral organocopper complex. The KIE experiments and molecular modeling supported the conclusion that the rate-limiting step of the reaction involved the transfer of the organic group from the copper center to the C4 position of the 2-methoxypyridine.

In an interesting study conducted by Harman and coworkers (Figure 4), the researchers analyzed the behavior of dihapto-coordinate 1,2-dihydropyridine complexes **13** of the metal fragment {WTp(NO)(PMe_3_)}(Tp = tris(pyrazolyl)borate), obtained from pyridine without any 1,4-dihydropyridine impurity [36]. The study revealed that these complexes undergo protonation at C6, followed by regioselective amination at C5 using various primary and secondary amines. Remarkably, the addition occurs stereoselectively *anti* to the metal center, leading to the exclusive formation of *cis*-disubstituted products. The resulting compounds, 1,2,5,6-tetrahydropyridines **18**, can be easily liberated through oxidation in acidic conditions. This tungsten-mediated procedure provides a means to access *cis*-2-substituted 5-amino-1,2,5,6-tetrahydropyridines **19** (also called 3-aminotetrahydropyridines), which are highly valuable scaffolds due to their presence in biologically active molecules. Notably, a wide range of nucleophiles, including Grignards, organozincs, enolates, indoles, and pyrroles can be added to C2, while the amine addition can be accomplished without the need for particularly expensive reagents, enabling control of the ring stereocenters. All these additions are regio- and stereoselective, yielding isolated products with a diastereomeric ratio greater than 15:1. Nevertheless, the reaction needs to be conducted at a low temperature to allow for the kinetic addition at C5 and the obtainment of the desired complex **18** avoiding an E1-elimination caused by the ammonium salt generated during the reaction which deprotonates complex **16** in C6 causing the formation of complex **13** that is thermodynamically favored. Moreover, to avoid the protonation-induced E1-type elimination a basic quench is required once the allylamine is formed, the subsequent oxidation with DDQ preconditioned with HOTf allows the desired product to be obtained. With this procedure it should be possible to access single enantiomers of the ATHPs based on the previously demonstrated enantioenrichment and stereochemical retention of {WTp(NO)(PMe_3_)} [37,38], which is applicable to pyridine-derived bicyclic amines [39]. 

In 2021, Smith and coworkers presented a novel asymmetric addition of alkynyl nucleophiles to *N*-alkylpyridinium electrophiles under mild conditions, obtaining dihydropyridine products with complete regiochemical and stereochemical control [40,41]. The pyridine derivative **20** was prepared via condensation between L-*tert*-leucine *tert*-butyl ester and 3-formylpyridine, followed by chemoselective methylation. Then, the treatment with propynyl magnesium bromide resulted in the selective formation of the desired dihydropyridine **21** in 70% yield, with complete stereoselectivity (>20:1 dr) and regioselectivity at C2 (>20:1 rr) (Figure 5). The asymmetric alkynylation of various substituted *N*-alkylpyridinium ions was investigated showing the broad scope and functional group tolerance of both the pyridine and the alkyne component. This method could be applied for the synthesis of a variety of dihydropyridines that can be broadly useful for the asymmetric synthesis of various azaheterocyclic building blocks. The proposed mechanism is based on what has been described by Koga for addition to unsaturated imines [37]; the chelation of the magnesium center by the imino ester guides the alkynyl nucleophile to the C2 position, a step that was also confirmed by X-ray analysis. The *tert*-butyl group putatively blocks the top face of the pyridinium, allowing for the alkyne nucleophile to approach from the opposite side, providing high stereocontrol. This method was then exploited for a concise, dearomative, and asymmetric synthesis of (+)-lupinine obtained in 75% yield with a >10:1 diastereomeric ratio with its epimer epilupinine.

A practical protocol for the regiodivergent asymmetric addition of aryl- and alkenyl-organometallic reagents to substituted *N*-alkyl pyridinium heterocycles was presented by Grigolo and Smith in 2022 [42]. The method allows for the achievement of high selectivity in both regiochemical and stereochemical control for the synthesis of chiral 1,2,3- and 1,3,4-trisubstituted dihydropyridine products. By carefully choosing the nitrogen activating agent it was possible to obtain regiocontrol and the authors discovered a direct relationship between the size of the activating group and the selectivity for C4 or C2 addition (activation with MeOTf favored addition in C2 with a 1:>20 ratio, while activation with trityl cation reversed the selectivity to favor addition in C4 with a 20:1 ratio) (Figure 6). The diastereoselectivities of the additions proved to be high for the major regioisomers obtained, suggesting that the chiral guide is effective regardless of the activating agent used. The authors also proposed a mechanism to explain the regio- and stereoselectivity of the reaction (Figure 6): in scenario **A**, a small activation group is used (e.g., Me), and it leads to the formation of intermediate **a** after adding the aryl Grignard reagent. The chiral guide has the *t*Bu group oriented on the opposite side to the chelated nucleophile, which allows for preferential addition to C2 of the pyridinium ring. In scenario **B**, a large activation group is employed (e.g., trityl), and when the nucleophile is added intermediate **b** is formed. This intermediate experiences a steric clash with the large activation group, causing the nucleophile to be controllably added to C4 of the pyridinium ring. In this second case the relative orientation of the chiral guide, pyridinium, and nucleophile lead to an inversion in facial selectivity and regioselectivity. The scope of additions at C2 was explored using allyl as a practical removable activating group, and various aryl and alkenyl Grignard reagents exhibited high regioselectivity and diastereoselectivity (>20:1) for each regioisomer formed. The researchers evaluated C4-selective asymmetric dearomatization too: activation of the pyridine derivative (**24**) with TIPSOTf was followed by reaction with aryl and alkenyl Grignard reagents and the subsequent LiOH workup yielded 1,4 adducts **25** with complete regiocontrol (>20:1 C4/C2) and diastereoselectivity (>20:1 dr). This controlled pyridinium functionalization was exploited in the total synthesis of complex molecular targets, exemplified by the synthesis of (+)-*N*-methylaspidospermidine and (+)-paroxetine. The protocol allowed for the avoidance of direct oxidative events, highlighting the strategic advantage of pyridinium functionalization in redox-economic alkaloid construction.

#### 2.1.2. Boronic Acids 

An interesting rhodium-catalyzed dearomatization of *N*-alkyl nicotinic acid esters for the synthesis of dihydropyridines that contain two double bonds with distinct reactivities showing a fully substituted stereocenter at the C6-position was reported in 2020 by Karimov [43]. The procedure involves the use of a nicotinic acid-ester-derived pyridinium salts in dioxane/H_2_O, with aryl boronic acids as nucleophiles, and the catalytic system is composed of Rh(COD)_2_BF_4_/(*R*)-BINAP (**L2**). These conditions create the desired 1,2-DHPs **27** with good yield and high stereoselectivity with the presence of KPF_6_ as an additive to improve the solubility of the pyridinium salt (Figure 7). The use of additives, ethereal solvents, and an amount of water in the reaction mixture are fundamental for the reaction yield and enantioselectivity. This protocol demonstrated excellent functional group tolerance for being compatible with a wide range of functional groups in the boronic acid. Interestingly, the alteration of the ester group, the substituent in C6, the *N*-alkyl group or the counterion in the pyridinium salt did not substantially influence the enantioselectivity of the reaction. Groups with high steric hindrance on the nitrogen atom reduced the ee’s slightly while the higher solubility of the pyridinium salt allowed higher yields of compounds of type **27**. The regioselectivity of the reaction was very high, no C4-addition was obtained and the ratio of C6/C2 addition was greater than 15:1, due to the higher activation of the C2 and C6 that are closer to the electro-withdrawing nitrogen atom. When using hindered substituents in C6, the major product derived from the addition of the boronic acid in C2 but the enantioselectivity was still maintained. Moreover, the enantioenriched dearomatization products can be selectively functionalized to prepare tetrahydropyridine and piperidine derivatives of type **28**. The DHPs were obtained through gram-scale synthesis with a maintained yield and enantioselectivity and then reduced using Pd/C, sodium cyanoborohydride, or LiAlH_4_ to obtain various enantioenriched tetrahydropyridines and piperidines with a fully substituted stereogenic center.

#### 2.1.3. Alkenes

In 2018, Buchwald and coworkers presented an interesting and peculiar copper-catalyzed asymmetric dearomatization of pyridines and pyridazines that allows C-C bond formation [44]. Within this method, the formation of the 1,4-DHP adduct **30** is only transient. The protocol involves mild reaction conditions, without requiring stoichiometric preactivation of the substrate, moreover, the nucleophile is formed in situ, and there is no need for protective group manipulation. The 1,4-DHPs are obtained by a catalytic high stereoselective 1,4-dearomatization and can be converted in substituted pyridines **31** (by oxidation, Figure 8 via a) or piperidines **32** (by reduction, Figure 8 via b) in the same pot. A wide variety of substituted pyridines and also pyridazines can be reacted with styrene in the presence of (Ph-BPE)CuH to obtain excellent conversions to the corresponding 1,4-DHP. The dearomatization showed high enantioselectivity and the aerobic aromatization proved to be stereospecific (Figure 8 via a). When reducing conditions are applied after the dearomatization step, good yields and selectivity are obtained even in gram scale. For the dearomatization/oxidation protocol for pyridines and pyridazines various functional groups were tolerated, but the ee’s of the pyridines were slightly reduced by electron-donating groups and enhanced by aryl and *π*-acceptor substituents. For the dearomatization/reduction protocol, only pyridines were tolerated, an analysis of the stereoselectivity of the reaction demonstrated that both steps are responsible for the control of the endocyclic stereocenter leading to a mixture of diastereomeric piperidines, and the major diastereomer was isolable in stereochemically pure form allowing for the obtainment of piperidines with three contiguous stereocenters starting from a prochiral precursor. The dearomative addition is selective for (C*α*,C4)-*anti* DHPs, whereas the reduction is selective for (C3, C4)-*syn* piperidines. Substituents at C3 are tolerated, while are accommodated at C4 only in particular cases; C2 substituents are not tolerated at all probably because the heterocycle needs to coordinate with a sterically demanding Lewis acid like copper. The procedure is applicable with variously substituted styrenes, except for *para*-F, -Me, and -OMe styrenes that completely impede dearomatization, due to possible destabilizing interactions in the dearomative transition state. 

As for the individuation of a mechanism for the reaction there has been a dispute between Buchwald’s group [44] initially proposing the cooperation of two copper(I) centers, and Lin’s group [45] proposing an elaborated FDT study in 2020, presenting a mechanism involving a single copper(I) center in which the reaction occurs through styrene insertion into Cu–H, to yield a Cu–H addition “product” that undergoes 1,3- and 1,5-Cu migrations on a benzylic ligand, facilitating the 1,2- and 1,4-dearomatization of pyridine. Lin’s group found that the comparable stability among the species presenting the benzylic ligand at different coordination sites allows for the Cu(I) migrations, promoting the C−C coupling between the benzylic carbon and a coordinated pyridine that completes the dearomatization process. From the calculations, Buchwald’s mechanism [44] involving two copper(I) centers was found to be unfavorable due to significant steric repulsions between the substituents on the ligands on the metal fragments, which hindered the dearomatization process. 

In 2020, Buchwald’s group addressed the reaction mechanism once again [46], proposing a monometallic process involving a dearomative rearrangement of the phenethylcopper nucleophile, leading to a C-*para*-metallated form before reacting with the heterocycle at position C4. The unexpected pathway is facilitated by the heterocycle itself, resulting in a net 1,5-Cu-migration and beginning with a doubly dearomative imidoyl-Cu-ene reaction, before the CuL_2_ fragment facilitates a stepwise Cope rearrangement of the doubly dearomatized intermediate, yielding the C4-functionalized 1,4-dihydropyridine (Figure 9). 

#### 2.1.4. Miscellaneous

The typical reactivity of pyridine or pyridinium salts permits the introduction of nucleophiles in the 2, 4, or 6 positions of the ring, generating 1,2- or 1,4-DHPs. In 2023, Fletcher and coworkers presented an innovative work that allows access to enantioenriched 3-substituted piperidines (**35**) starting from pyridine and sp^2^-hybridized boronic acids [47] (Figure 10). In this case, the protocol consists of two steps: the first is the formation of 1,2-DHP (**34**) via the selective Fowler reduction of an in situ-generated pyridinium salt, and the second is a Rh-catalyzed carbometallation of the obtained 1,2-DHP. After a screening of activating groups for the formation of the pyridinium ion, the author found that phenyl carbamate-protected DHPs were the best performing both for the high yield that was obtained and the facile purification of the solid DHP. Fletcher’s group then focused on the optimization of the procedure for the Rh-catalyzed carbometallation of phenyl carbamate dihydropyridines. The optimized protocol involves the use of [Rh(cod)(OH)]_2_, Segphos (**L_5_**), aqueous CsOH in THP:toluene:H_2_O (1:1:1) mixture at 70 °C, these conditions allowed for the obtainment of tetrahydropyridines in high yields and with very high ee’s. The use of aqueous CsOH is necessary for the reaction, as it is the source of water, rhodium, and ligand. The scope of the boronic acids was extensively analyzed, and a broad range of boronic acids with numerous functional groups and heterocyclic moiety were easily tolerated, maintaining good yields and enantioselectivity. The carbamate activating group can be modified as well, allowing for the introduction of groups that can be easily modified for the synthesis of specific compounds; alkyl carbamate groups were tolerated, maintaining good results. This reactivity was also maintained with 2-substituted pyridines and variously substituted dihydroquinolines, while 4-substituted dihydropyridines and azepine were not reactive. This synthetic method allows for the synthesis of enantioenriched tetrahydropyridines from 1,2-dihydropyridines, which can be performed on gram scale and can be used for the synthesis of enantioenriched piperidines which are important scaffolds in medicinal chemistry. 

### 2.2. Organocatalysis 

Whereas the use of metal-catalyzed reactions for the dearomatization of pyridine has a long history in the literature, organocatalyzed reactions gained chemists’ interest only recently because of the ready availability, low cost, low toxicity, and high variability of the catalysts that can be exploited [25,26,27,28,29]. 

Fogagnolo and coworkers in 2018 reported an interesting study on the chiral *N*-heterocyclic carbene (NHC)-catalyzed intermolecular dearomatization reaction between activated *N*-alkylpyridinium salts and aliphatic aldehydes to obtain acylated 1,4-dihydropyridines with complete C4-regioselectivity and enantioselectivities ranging from 52% to 78% ee’s [28]. The protocol is based on the obtainment of 1,4-DHP of type **37** with complete regioselectivity, high yield, and promising enantiomeric excess through the use of a *N*-benzylpyridinium salt containing a cyano group at the 3-position, a catalytic amount (10 mol %) of amino-indanol-derived triazolium salt **C_1_**, and K_3_PO_4_, or better Na_2_CO_3_, in toluene (Figure 11). The authors noticed that the increase of the polarity of the reaction medium reduced reaction efficiency, but the use of apolar CCl_4_ restored enantiocontrol by the catalyst (74% ee), even if giving a diminished yield likely due to a lower solubility of the pyridinium salt. As for the base, K_3_PO_4_ and Na_2_CO_3_ could be used alternatively and exhibited better enantioselectivity compared to Cs_2_CO_3_, probably due to a correlation between the hard/soft character of the metal and the stereochemical outcome of the dearomatization process. The study revealed that the conformational freedom of the alkyl substituent on the aldehyde in the transition state can influence the enantioselectivity (less restricted conformational freedom in the transition state results in lower ee’s). The methodology exhibited high selectivities, and usually the moderate yields obtained for certain 1,4-DHPs were associated with low conversions of the substrates. The limitations of the protocol were investigated as well by the authors, such as the ineffectiveness of variations in the electron-withdrawing group at the C3 position of the *N*-benzylpyridinium ring and the reduced efficacy of aromatic aldehydes in the dearomatization process. The proposed mechanism consists of the NHC **i** (obtained by deprotonation of triazolium salt **C1**) that reacts with the aldehyde to give the intermediate **ii**, which intercepts the pyridinium salt **36** to give the adduct **iii**. The deprotonation given by the base creates product **37** and regenerates the catalyst. This methodology enabled the synthesis of previously unreported C4-acylated 1,4-DHPs **30** with a carbonyl functionality, two enamine-type double bonds, and a cyanide group. The 1,4-DHP scaffold could be variously elaborated by chemoselective reductions with NaBH_4_ and H_2_, Pd(OH)_2_ to give 1,4-DHPs **38** and tetrahydropyridines **39**, respectively. 

In the same year, Chen and coworkers presented an asymmetric dearomative formal [4 + 2] cycloaddition reaction of activated *N*,4-dialkylpyridinium salts **40** and acyclic *α*,*β*-unsaturated ketones **41** via the ion cascade iminium-enamine catalysis of a cinchona-derived amine **C_2_** [48]. The authors envisioned that in mild basic conditions the iminium ion-enamine tautomerization of the alkylpyridinium salts would generate dienamine-type intermediates, which are enantioselectively trapped by *α,β*-unsaturated ketone substrates activated by the formation of iminium ions with a chiral amine. The intramolecular dearomative reaction then allows the formation of chiral azaspiro [5.5]undecane derivatives **42** in a formal [4 + 2] cycloaddition pattern. The optimized reaction conditions for the dearomative formal [4 + 2] reaction involve the activated pyridinium substrate with a 3-nitro group and the enone in CHCl_3_ at 60 °C in the presence of chiral primary amine **C_2_** (20 mol%), a substituted mandelic acid **A1** (40 mol%), and sodium acetate (1.2 equiv) obtaining excellent yield (94%), diastereoselectivity (>19:1 dr), and enantioselectivity (95% ee) (Figure 12). In general, good to excellent diastereo- and enantioselectivity were obtained for *β*-aryl enones with diverse electron-withdrawing or electron-donating groups, but also for enones bearing a heteroaryl or 2-styryl group, while enones with a linear alkyl group obtained lower enantioselectivity. Various substitutions for the pyridinium salt were evaluated, the *N*-substitution did not affect the selectivity, and various groups were tolerated with only few exceptions. The obtained dearomative product is also suitable for further transformations, such as full hydrogenation with Pd/C and H_2_, deprotection of the *N*-benzyl group, and protection with Boc using (Boc)_2_O to create **43**, or ring-opening reactions with BF_3_·Et_2_O in CH_2_Cl_2_ producing **44**. This interesting work presents a new type of bench-stable activated N,4-dialkylpyridinium salts that can be easily deprotonated to generate active dearomative dienamine-type intermediates and participate in an asymmetric formal [4 + 2] cycloaddition reaction with acyclic ion *α,β*-unsaturated ketones through the cascade iminium-enamine catalysis of a cinchona-derived amine with moderate to excellent stereoselectivity.

In 2020, following on from their previous work Chen reported an enantioselective cascade reaction for the synthesis of fused polyheterocycles using *N*-alkylpyridinium and *N*-alkylquinolinium salts **45** with o-hydroxybenzylidene acetones **37** [49]. The authors also developed a cascade assembly using *N*-benzyl-4-methylpyridinium salt and cyclic 2,4-dienones involving repetitive dearomatization and aromatization activation and resulting in bridged frameworks. The reactions proceeded through dearomative dienamine-mediated addition, consecutive trapping of reactive enamine intermediates, and aminal formation under the catalysis of cinchona-derived primary amines. Moreover, the researchers performed multiple functionalizations of 4-methylpyridinium salts with cyclic 2,4-dienone substrates, resulting in bridged and fused frameworks via a domino regioselective Michael/Michael/Mannich sequence with moderate to good enantioselectivity. The optimized reaction conditions are with the use of *N*-benzyl-3-cyanopyridinium **45** salt and *o*-hydroxybenzylideneacetone **46**. The reaction, catalyzed by quinine-derived primary amine **C_2_** (20 mol%) and salicylic acid **A_2_** (20 mol%) in the presence of potassium salicylate (1.1 equiv), created the desired product in 78% yield and 98% ee as a single diastereomer (Figure 13). The aminal formation is crucial for the cascade process that does not occur for simple benzylideneacetone. Pyridinium salts with diverse *N*-benzyl or other *N*-alkyl substituents were well-tolerated with this protocol, but without a 3-cyano group only complex mixtures were obtained. The products with the opposite configuration were produced with similar good data obtained by employing amine with the opposite configuration. This research demonstrated the utility of activated pyridinium and quinolinium salts in asymmetric dearomative multiple-functionalization reactions with o-hydroxybenzylideneacetones. 

The addition of a C(1)-ammonium enolate intermediate to the pyridinium salts for the enantioselective synthesis of *α*-functionalized ester-substituted 1,4-DHPs was reported in 2021 by Smith and coworkers [50]. The C(1)-ammonium enolate intermediate is formed in situ from aryl esters and the isothiourea catalyst (*R*)-BTM **C3**, giving a regio- and stereoselective formation of enantioenriched 1,4-DHPs **50** (Figure 14). Extensive screening of reaction conditions allowed the authors to understand the importance of using toluene as the solvent of the reaction, as the low solubility of the pyridinium salt in toluene limited background reactions leading to the formation of a racemate giving a high stereocontrol. The base obtaining the best results is DABCO (65% yield, 90:10 dr, 91:9 er) and it is necessary to neutralize the HBr generated during the reaction and reduce the deactivation of the isothiourea catalyst by protonation. Moreover, the counter ion of the pyridinium salt proved to be important for the reactivity. Coordinating counter ions such as Br^−^ and Cl^−^ appeared to be compatible with the catalytic protocol, due to their binding to the isothiourea stabilizing the transition state by non-covalent interactions, while larger and non-coordinating counter ions such as BF_4_ and PF_6_ gave lower yields and enantioselectivities. Furthermore, for this reactivity an electron-withdrawing substituent (nitro, nitrile, acetyl, or 3-phenylsulphonyl) in the 3-position of the pyridinium salt was necessary and an *N*-benzyl-derived substituent was essential as well, probably due to stabilizing π–cation or ππ-interactions in the transition state involving the benzyl group. Various aryl acetic *p*-nitrophenyl ester components with different steric and electronic properties were also evaluated and obtained good yields, diastereo-, and enantiocontrol, with only a few exceptions such as strongly electron-withdrawing substituents that make the ester acidic and give stronger racemic background reactions. After the catalytic process, various nucleophiles (especially amines) were added to the reaction mixture generating numerous carbonyl moieties. Mechanistic studies were also performed in this work, showing that stereocontrol can derive from the ability of the pyridinium salt to interact with donor ππ-systems given by the aryl acetic ester through π-cation-interactions and form a pre-transition state with lower steric hindrance. This interesting work broadens the scope of isothiourea catalysis which had been limited to alkenes and carbonyl derivatives.

In 2022, Feng and Cao reported an innovative catalytic asymmetric sequential three-component nucleophilic addition/dearomative [4 + 2] cycloaddition/isomerization cascade reaction which involves the use of a methylene-indolinones electrophile **51** that traps the transient zwitterions formed by *N*-heteroaromatic compounds, and allenoates **53** and **56** [51]. This multi-component dearomatization strategy is highly efficient for the synthesis of structures presenting molecular complexity and diversity, with impressive yields reaching up to 92%, diastereomeric ratios higher than 19:1, and enantiomeric excesses up to 99%. It is performed in CH_2_Cl_2_ at 20 °C in the presence of a base like Et_3_N that accelerates the isomerization and water that reduces background reactions, increasing yield and enantioselectivity (Figure 15). The reaction scope encompassed various structures such as 1,2-dihydroquinolines, 1,2-dihydropyridines, and other *N*-heterocycles. Notably, the enantioselectivities and diastereoselectivities of 1,2-dihydroisoquinoline derivatives remained consistent even upon altering the ester group, and pyridine maintained a similar reactivity; only changing the chiral *N,N*′-dioxide ligand from **L_4_**-PicH to **L_4_**-PrEt_2_Me and prolonging the reaction time without adding the base achieved the corresponding products **54** and **57** with good yields and high enantioselectivities, as a single diastereomer. The scope of substituents for the synthesis of 1,2-DHPs was investigated by the authors: C4 and C3 substituents on the pyridines allowed high selectivity with C4-electron-withdrawing groups, increasing enantioselectivities, while for 3-substituted pyridines, regioselectivity was influenced by the steric hindrance and electronic effects of the substituents. The dearomative [4 + 2] cycloaddition between methyleneindolinone and the generated zwitterion exhibited very good diastereoselectivity. Both gram-scale synthesis and derivatization of the obtained compounds were achieved with good results. An analysis of the mechanism of the reaction was performed using deuterium labeling, allowing for an understanding of the regio- and stereoselectivity of the reaction and the role of the additives. With this reactivity, it is possible to obtain various chiral polycyclic *N*-heteroaromatic compounds and complex chiral molecules.

In 2022, Reisman and coworkers presented an unprecedented bioinspired dearomative annulation between pyridine and glutaryl chloride that resulted in the first total synthesis of the lupin alkaloid (−)-sophoridine (**69**), and the shortest syntheses of other alkaloids such as (+)-isomatrine (**66**), (+)-matrine (**71**), (+)-allomatrine (**70**), and (+)-isosophoridine (**68**) to date [52]. The (+)-Matrine (**71**) and (+)-isomatrine (**66**) are tetracyclic alkaloids suggested to derive from three molecules of (−)-lysine via the intermediacy of the unstable cyclic imine Δ1-piperidine (Figure 16). Reisman et al. envisioned that pyridine could serve as a stable, inexpensive synthon for Δ1-piperidine, and the remaining five carbons of the tetracyclic matrine framework could derive from glutaryl chloride **58**, proposing a dearomative annulation via a bis-acylpyridinium salt to form a tetracycle. The reaction between the pyridine and glutaryl chloride in CH_2_Cl_2_ yielded the (±)-tetracycle **64** with a 62% yield on a 10 g scale (a one mole scale reaction was also performed). The (±)-tetracycle **64** was subjected to single-crystal X-ray diffraction confirming the *syn-syn* relative stereochemistry. The ^1^H NMR analysis showed a possible equilibrium between bis-acylpyridinium salt **59** (dominant at −40 °C) and the acid chloride resulting from monocyclization **62** (dominant at 25 °C). Computational investigations revealed that the lowest-energy transition state for the first cyclization step involved a boat-like conformation to form the *syn* isomer. The final deprotonation step was found to be the selectivity-determining step, favoring the formation of *syn-syn* (±)-tetracycle **64** over the *anti-anti*, despite being thermodynamically less stable. This result was consistent with the experimental observation of (±)-tetracycle as the sole diastereomer, despite the initial mixture of monocyclization products. Hydrogenation of (±)-tetracycle **64** and the following reduction with alane generated (±)-isomatridine (**65**) in 60% yield over two steps. The resolution of diamine (±)-isomatridine **65** was performed via recrystallization of the di-*p*-toluoyl tartaric acid salt (24% recovery, 90% ee) of the (+)-diamine isomatridine. The isomerization of (+)-isomatrine (**66**) was achieved with Rh/C obtaining (+)-matrine (**71**) (32% yield), while (+)-allomatrine (**70**) was obtained in 83% yield with Pd/C. Isomerization with Pt/C obtained (+)-isosophoridine (**68**) in 55% yield. The use of PtO_2_ at 98 °C for 15 min allowed for the obtainment of (−)-sophoridine (**69**) in 10% yield. When the reaction with PtO_2_ was conducted at 80 °C for 24 h, the (−)-isomer of an unnatural product (**67**) was isolated in 40% yield (Figure 13).

Another extremely peculiar method for the asymmetric dearomatization of activated pyridines for the preparation of stereo-defined 3- and 3,4-substituted piperidines is presented by Turner and coworkers [53]. This chemo-enzymatic approach exploits a stereoselective one-pot amine oxidase/ene imine reductase cascade combining the mild chemical reduction of pyridiniums to tetrahydropyridines (THPs) with the stereoselectivity of a biocatalytic cascade to reduce the C=C bond. The biocatalytic oxidation with an amine oxidase (AmOx) of the THP (**73**) in situ generates the corresponding dihydropyridiniums (DHPs) (**74**), creating an activated C=C bond conjugated to the C=N bond, which is then reduced with the biocatalyst to generate a cascade reaction to obtain piperidines (**75**) (Figure 17). The conversion of a series of substituted *N*-alkyl THPs (obtained by the reduction of activated pyridines (**72**) with NaBH_4_) into piperidines using AmOxs in combination with EREDs or EneIREDs was screened; the authors found that the 6-hydroxy-D-nicotine oxidase (6-HDNO) variant, E350L/E352D, is effective with a broad substrate scope in the oxidation step. For the reduction of the C=C bond of the α,β-unsaturated iminium ion, the EneIRED from an unidentified Pseudomonas sp., in combination with 6-HDNO, reduced a wide number of THPs generating piperidines in good yield and with excellent enantioselectivity. By screening the metagenomic IRED collection, in combination with the 6-HDNO variant, the authors identified the biocatalysts capable of generating either enantiomer of piperidine from the THP allowing for the division of EneIREDs into two groups depending on the chirality of the obtained piperidines. Enzymes in both series showed a broad tolerance for aryl substituents at the C-3-position of the THP scaffold, moreover, a variety of *N*-alkyl substituents were accepted. Even hindered 3,4-disubstitituted THPs and a combination of alkyl and aryl substituents at C-3 and C-4 could be reduced with the formation of *cys* (**75b**) or *trans* (**75a**) piperidines dependent on the kind of substituents on the scaffold. This cascade reaction was applied to the synthesis of a precursor of Niraparib, a PARP inhibitor for the treatment of ovarian cancer. 

## 3. Dearomative Substitution at Nucleophilic Pyridine Nitrogen

The stereoselective dearomatization of the pyridine nucleus can also occur by means of pyridine derivatives that behave as nucleophiles at the nitrogen. This kind of approach, that has come to the fore more recently, can be declined in several ways. A fundamental tract of this reactivity relies on the use of 2-substituted pyridine derivatives and often allows straightforward access to *N*-substituted 2-pyridones, which are important structural motifs found in natural products and APIs such as pirfenidone, doravirine and palbociclib [8,9,54,55,56,57,58]. Traditional methods for the synthesis of *N*-substituted pyridones are plagued by the ambident nucleophilic nature of 2-pyridones. Despite the recent considerable progress, the selective *N*-alkylation is still problematic especially when it is necessary to introduce a branched alkyl chain comprising a chiral center α to the nitrogen as is present in compounds shown in Figure 2 which possess interesting pharmacological properties. The synthesis of these compounds has so far relied upon the formation of a pyridine ring starting from chiral amines. 

Recently, several elegant direct approaches to obtaining chiral 2-pyridones in an asymmetric fashion have been developed featuring intramolecular and intermolecular reactivity generally starting from 2-substituted pyridines. 

### 3.1. Intramolecular Dearomative Reactions of 2-Substituted Pyridines

In a seminal paper, Batey and coworkers reported an enantioselective palladium(II)-catalyzed formal [3.3]-sigmatropic rearrangement strategy of 2-allyloxypyridines **76** [59]. In the optimized reaction conditions, planar chiral Pd(II) complex (*S*)-COP-Cl (5 mol%) and AgOCOCF_3_ (10 mol%) at 45 °C in DCM were applied toward the rearrangement of a variety of allyloxypyridines **76** that can be easily obtained by microwave-assisted nucleophilic aromatic substitution reactions (Figure 18). Both (*E*)- and (*Z*)-pyridine substrates rearranged to the corresponding chiral 2-pyridones **77** with good yields and enantioselectivity up to 90%. The only limitation was given by particularly hindered substrates (R = Chx, no reaction) or (R = Ph, 30% yield).

On the other hand, suprafacial [1,3]-sigmatropic rearrangements that are thermally disallowed can be promoted starting from 2-benzyloxypyridine derivatives using Ru [60] and Ir catalysts [61] at elevated temperatures (80–135 °C). A very particular enantioselective *O*- to *N*-[1,3]-rearrangement occurring at temperature as low as −40 °C has been reported by Cordier and coworkers, making use of chiral copper complexes of diphosphine as catalysts (Figure 19) [62]. In this way, starting from racemic 2-propargyloxypyridines **78** it is possible to obtain enantioenriched *N*-propargylic 2-pyridones **79** with ee’s up to 95%. In depth mechanistic investigations showed the likely intervention of bimetallic copper-acetylides in which coordination to the pyridyl nitrogen represents a crucial interaction. From a synthetic perspective, an important limitation is given by the requirement of two nitro substituents on the starting pyridine.

Another way to exploit the nucleophilic character of the pyridine nitrogen in an intramolecular dearomative stereoselective fashion relies on the presence of an allylic leaving group at a suitable distance. You and coworkers reported a direct asymmetric dearomatization of pyridine derivatives and related heterocycles (pyrazines, quinolines, isoquinolines) by an iridium-catalyzed allylic amination reaction of a pendant allylic carbonate [63,64]. Key to success are: (i) the iridium-catalyzed oxidative addition to the allylic carbonate (ii) the presence of an EWG (ester or ketone in the R^2^ position) then the acidic H_a_ is deprotonated by the liberated methoxy anion, as shown in Figure 20. 

By employing this method, a series of 2,3-dihydroindolizines **80** were easily prepared in high yields and enantioselectivities.

The catalytic asymmetric dearomatization of (nucleophilic) pyridine derivatives have been recently used for an unprecedented intramolecular atroposelective cycloisomerization to axially chiral arylquinolizones **82** (Figure 21) [65]. A positive interplay between the chiral copper-phosphoramidite catalyst **L_5_** and a chiral Brønsted acid such as (*R*)-CPA **L_6_** was found in the optimized reaction conditions. From the mechanistic study made by the authors it seems that the copper-catalyst plays a role in controlling the stereoselectivity while the Brønsted acid is necessary to promote the reactivity via coordination to the oxygen of the carbonyl group. 

### 3.2. Intermolecular Dearomative Reactions of (2-Hydroxy)Pyridines

The intermolecular reactivity of 2-hydroxypyridine is complicated by the existence of a tautomeric equilibrium with 2-pyridone. Therefore, the *N* vs. *O* selectivity of the electrophilic alkylation of this equilibrating nucleophile can be fundamentally problematic. Moreover, pyridones are known as good ligands for transition metal catalysts [66] making transition-metal-catalyzed asymmetric amination more challenging. In a seminal paper in this field, Breit and coworkers demonstrated the prominent role of the more acidic 2-hydroxypyridine tautomer undergoing a rhodium-catalyzed addition to allenes under neutral conditions to initially give a kinetic *O*-allylation product **83** that was finally converted into the thermodynamically more stable *N*-allyl 2-pyridone **84** in high yield and ee’s up to 98% (Figure 22) [67]. 

Soon after, the You group reported an iridium-catalyzed intermolecular asymmetric allylic amination of 2-hydroxypyridine with allylic carbonates [68]. Importantly, the reaction did not occur at all in the absence of a base, with Cs_2_CO_3_ (40 mol%) showing the best results. The best substrates were aryl allylic carbonates whereas aliphatic allylic carbonates were much less selective in terms of chemoselectivity and regioselectivity. For example, for R^2^ = *n*Pr a *N*/*O* = 86/14 and a regioselectivity ratio of 75/25 between branched and linear products of type **77** and **85** were achieved (Figure 23).

A further evolution of this chemistry is based on in situ isomerization of the enantioenriched allylic product by an organic base such as DBU to give axially chiral enamides via a “one-pot” two-steps process [69].

In order to overcome the natural tendency of palladium-catalyzed allylic substitution reactions to give linear achiral products, Zhang and coworkers exploited the hydrogen-bond interactions between nucleophilic 2-hydroxypyridine and allylpalladium intermediates [70]. In the optimized reaction conditions, hydroxyl-containing allylic carbonates were allowed to react with 2-hydroxypyridines in the presence of palladium complexes with Feringa’s phosphoramidite **L_6_** to give *N*-substituted 2-pyridones **86** with complete chemo- and regioselectivity and good to high enantioselectivities. A poor branch selectivity was obtained when using methoxy allylic carbonate or 4-hydroxypyridine (Figure 24). The authors also demonstrated that initial *O*-alkylation followed by rearrangement (as previously described for the related Rh-catalyzed procedure) is unlikely to be involved.

A related approach using vinyl cyclic carbonates as the electrophilic partner was developed subsequently by Khan and coworkers [71]. As allylic carbonates had already been described as competent electrophiles capable of directing reactivity via secondary hydrogen-bond interactions [70], the real novelty of this article is the possibility of obtaining a variety of tertiary allylic-2-pyridones **87** containing a quaternary stereogenic center with moderate to good yields and very high enantioselectivities. The best results were this time achieved using Pd(0)/DACH-naphthyl Trost-type ligand **L_7_** (Figure 25). 

A very particular approach to obtaining *α*-substituted chiral substituted piperidines and tetrahydropyridines in non-racemic form makes use of the pyridine quaternary salts formed in situ from pyridines and haloacetamides as nucleophiles (Figure 26) [72]. A palladium-catalyzed allylic alkylation using allylic carbonates in the presence of a catalytic amount (6 mol%) of chiral ferrocenyl phosphine ligand **L_8_** generated the corresponding pyridinium salt **88** with high yields and enantioselectivities. The subsequent dearomatizing reduction can be carried out with Raney Ni to give the corresponding chiral piperidines **89** or with NaBH_4_ resulting in chiral tetrahydropyridines **90**. 

The inherent preferential for *O*- over *N*-reactivity of 2-hydroxypyridines with diazo compounds can be overcome by the use of 2-*O*-substituted pyridines such as *O*-Boc pyridine in the presence of rhodium catalysts. In this way, the in situ formation of pyridinium ylide of the general structure **91** followed by the 1,4-acyl rearrangement deliver *N*-substituted 2-pyridones **92** (Figure 27a). Based on these results, an asymmetric version of the reaction was developed by the use of 1 mol% of Rh_2_(S-TCPTTL)_4_ **L_9_** on a range of substrates. Beyond Boc as the R^2^ group, the amide group was also applicable to the reaction with slightly lower yields and enantioselectivities (Figure 27b) [73].

As this protocol was limited to stabilized diazo compounds bearing EWG groups, the same research group also reported a chemo- and enantioselective insertion of furyl carbenes generated in situ from readily available enynones using the same chiral dirhodium complexes (Figure 28) [74].

The authors showed by means of DFT calculations that the reaction proceeded through enantioselective pyridinium ylide formation and sequential 1,4-proton transfer with steric repulsion and *π-π* interaction in the catalyst pocket accounting for the chemo- and enantioselectivity.

In a completely different approach, chiral *N*-substituted 2-pyridones of type **94** were prepared by an enantioselective aza-Michael addition of halogenated 2-hydroxypyridines to *α,β*-unsaturated-1,4-diketones or 1,4-ketoester catalyzed by squaramide cinchona alkaloids such as **C_4_** [75]. The main limitations of this method are given by the necessary use of halogenated 2-hydroxypyridines and by the (*E*)-stereochemistry of the Michael acceptor in order to obtain high yields and enantioselectivities (Figure 29).

### 3.3. Intermolecular Reactions of 4-Hydroxypyridines

4-Hydropyridines and tautomeric 4-pyridones are versatile building blocks for the synthesis of alkaloid natural products and medicinally relevant molecules. In particular, direct asymmetric synthetic routes to *N*-alkylated 4-pyridones bearing a stereocenter α to the nitrogen atom have not been much explored mainly due competing *N*- and *O*-alkylation processes and their low nucleophilicity and chelating properties. You and coworkers recently reported a copper-catalyzed intermolecular propargylic amination of 4-hydroxypyridines to give *N*-alkylated derivatives **95** (Figure 30) [76]. 

The authors rationalize the stereocontrol of the asymmetric process with an edge-to-face interaction between the copper allenylidene and fluorobenzene ring of the Pybox ligand **L_10_**. The main limitations of the methods are related to the non-reactivity of the aliphatic propargylic substrate and very long reaction times at the cryogenic conditions (−40 °C) necessary to obtain high enantioselectivity.

Building on previous observations (Figure 22), Breit and coworkers extended the rhodium-catalyzed addition to terminal allenes using 4-hydroxypyridines as the nucleophilic partner. The reaction showed complete chemoselectivity towards *N*-allylated pyridones of type **96** which were obtained with high enantioselectivities although the regioselectivity (branch/linear) was not complete (Figure 31) [77]. Switching the catalyst to a Pd/dppf [dppf = 1,1′-bis(diphenylphosphino)ferrocene] system determined a complete reversal of regioselectivity in favor of the achiral linear *E*-alkene isomer of type **97**.

A noteworthy extension of the allylic partner to allylic alcohols in asymmetric allylic amination with pyridines was described by You and coworkers. The catalyst system was made by an iridium complex with Carreira’s [P/olefin] ligand **L_12_** with the necessary inclusion of a Lewis acid such as Zn(OTf)_2_. The reaction obtained the corresponding allylated pyridines of type **98** with excellent chemo-, regio- and enantioselectivities (>19:1 N/O, >19/1 branch/linear and ≥89% ee) (Figure 32) [78].

## 4. Conclusions

The possibility of building up complex molecules starting from easily available pyridines via the dearomatization reaction has come to the fore in recent years as a particularly versatile approach. In this article we have summarized the catalytic and stereoselective synthesis of partially hydrogenated pyridines and of pyridones via the dearomative reactions of pyridine derivatives up to mid-2023. Traditionally, due to its scarce reactivity, activation of the pyridine nucleus by *N*-functionalization is required to efficiently perform a nucleophilic addition. As regards this more frequently encountered approach, our paper complements and updates previous review articles published in the last decade [4,10,11]. On the other hand, we also report a collection of examples based on a less explored complementary strategy where the stereoselective dearomatization occurs by means of pyridine derivatives that behave as nucleophiles at the nitrogen site. So far, this more recent strategy has been mostly developed on 2-substituted pyridine derivatives leveraging inter- and intra-molecular allylic amination reactions or sigmatropic rearrangements to create chiral 2-pyridones. 

Overall, the hydropyridines obtained through dearomatization reactions are valuable products in consideration of the role that such heterocyclic derivatives have in medicinal drugs. Moreover, the further elaboration of hydropyridines in particular by cycloaddition reactions can offer the possibility to prepare novel stereochemically complex sp^3^-rich azaheterocyclic scaffolds, creating new inputs and solutions in medicinal chemistry programs aimed at the discovery of new bioactive molecules and novel mechanisms of action.

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
