# Peer review of "Recent Progresses in the Catalytic Stereoselective Dearomatization of Pyridines"

_molecules, 2023, doi:10.3390/molecules28176186_

Round 1

Reviewer 1 Report

The authors reviewed recent progress in the dearomatization of pyridine derivatives. Numerous papers are cited, covering the field. However, there are many problems that need to be considered in order to be accepted as a review article.

In general, there are lack of Schemes, and many parts cannot be understood without reading the original documents. If the discussion is to be included in the text, the Scheme should also be provided. In addition, the relationship between each of the cited papers is unclear, just by listing the results of original documents. It is a review article that is difficult for the reader to understand. The reviewer suggests a major revision of the composition.

Other minor points:

1.      The order in which the compounds are presented in the text should be consistent with the Scheme; in Scheme 1, pyridine should be placed in the upper row and the compound number of pyridine should be 1.

2.      The Citation numbers are incorrect (for example, 27 in the text should be 25 in reference section) and all of them should be confirmed by checking again.

3.      P3, line 82; the word ‘1,4-derivative derivative’ should be changed.

4.      P3, Scheme 1; The structures of dihydroquinoline 2 and tetrahydroquinoline 3 are incorrect (benzene ring).

5.      P4, Scheme 2; In the original paper, the authors used various Grignard reagents, which should be described as R'MgBr in order to describe other reagents than EtMgBr.

6.      In the dearomatization of pyridine derivatives, the reaction mechanism is important, and the reaction mechanism should be given in this paper for representative examples (although not necessarily all of them). For example, the regioselectivity of Scheme 6 cannot be understood by only textual explanation.

7.      P6, Scheme 5; R1-X should be denoted as the activator.

8.      P6, Scheme 5; The title of Scheme 5 is ‘… alkynylation of N-alkyl pyridinium…’, however, pyridinium is an intermediate and is not shown in the Scheme. The title should be changed.

9.      P6, line 195; compound 22 and 24 are not N-alkyl pyridinium.

10.  As for the sectioning of the paper, 2.1 and 2 refer to nucleophiles, 2.3 to organocatalysts, and 2.4 to alkenes, these are not on an equal footing. There should be a large classification into metal catalysts and organocatalysts, and the differences by nucleophile should be indicated under them.

11.  P8, line 284, 288, 297 and many more; garbled characters

12.  P9. Scheme 9; Further transformations are mentioned in the text, but are not described in the Scheme, so the reader cannot understand them.

13.  Figure 2 shows examples of N-substituted pyridone derivatives, however, are these compounds synthesized by dearomatization? If not, this is a statement that is not necessary in a review article (it only shows that they can be synthesized by other methods, not the data that strengthens the importance of dearomatization).

This manuscript should be accepted after these concerns have been addressed.

Author Response

The authors reviewed recent progress in the dearomatization of pyridine derivatives. Numerous papers are cited, covering the field. However, there are many problems that need to be considered in order to be accepted as a review article.

In general, there are lack of Schemes, and many parts cannot be understood without reading the original documents. If the discussion is to be included in the text, the Scheme should also be provided. In addition, the relationship between each of the cited papers is unclear, just by listing the results of original documents. It is a review article that is difficult for the reader to understand. The reviewer suggests a major revision of the composition.

OUR REPLY: We thank the Reviewer for this general comment. The overall material present in our submission has been clearly divided into sections in accordance with the clear logic and indications displayed in the text. As it is normal within a review article, we have summarized all the contributions in the field and therefore detailed explanations and Schemes cannot be provided. We think that the descriptions provided are enough to understand what has been done, otherwise it is normal that to go into deeper details the reader should dip into original literature. Anyway, we have now introduced some more explanatory details in Schemes 1,3,5,6,11 and 13 and all minor points listed below:

  1. The order in which the compounds are presented in the text should be consistent with the Scheme; in Scheme 1, pyridine should be placed in the upper row and the compound number of pyridine should be 1. (OUR REPLY: Pyridine was numbered 1)
  2. The Citation numbers are incorrect (for example, 27 in the text should be 25 in reference section) and all of them should be confirmed by checking again. (OUR REPLY: all citations have been corrected. We thank the Reviewer to bring this important issue to our attention)
  3. P3, line 82; the word ‘1,4-derivative derivative’ should be changed. (corrected)
  4. P3, Scheme 1; The structures of dihydroquinoline 2 and tetrahydroquinoline 3 are incorrect (benzene ring). (corrected)
  5. P4, Scheme 2; In the original paper, the authors used various Grignard reagents, which should be described as R'MgBr in order to describe other reagents than EtMgBr. (R’MgBr was used)
  6. In the dearomatization of pyridine derivatives, the reaction mechanism is important, and the reaction mechanism should be given in this paper for representative examples (although not necessarily all of them). For example, the regioselectivity of Scheme 6 cannot be understood by only textual explanation. (some mechanisms have been presented both in the text and in the schemes. Check Scheme 1 and text, Scheme 3 and text lines 133-138, Scheme 5 and text lines 186-191, Scheme 6 and text lines 206-215, Scheme 11 and text lines 288-292, Scheme 13 and text)
  7. P6, Scheme 5; R1-X should be denoted as the activator. (done)
  8. P6, Scheme 5; The title of Scheme 5 is ‘… alkynylation of N-alkyl pyridinium…’, however, pyridinium is an intermediate and is not shown in the Scheme. The title should be changed. (title has been changed)
  9. P6, line 195; compounds 22 and 24 are not N-alkyl pyridinium. (the definition of the compounds has been changed)
  10. As for the sectioning of the paper, 2.1 and 2 refer to nucleophiles, 2.3 to organocatalysts, and 2.4 to alkenes, these are not on an equal footing. There should be a large classification into metal catalysts and organocatalysts, and the differences by nucleophile should be indicated under them. (the changes proposed by the referee have been done, now the Nucleophile part is divided in metal catalysts and organocatalysts)
  11. P8, line 284, 288, 297 and many more; garbled characters (Greek letters have been rewritten in place of the “garbled” characters, probably a problem in the transfer of the files)
  12. P9. Scheme 9; Further transformations are mentioned in the text, but are not described in the Scheme, so the reader cannot understand them. (the transformations have been presented in the Scheme 12, former Scheme 9)
  13. Figure 2 shows examples of N-substituted pyridone derivatives, however, are these compounds synthesized by dearomatization? If not, this is a statement that is not necessary in a review article (it only shows that they can be synthesized by other methods, not the data that strengthens the importance of dearomatization). It has been clearly written that: The synthesis of these compounds has so far relied upon the formation of pyridine ring starting from chiral amines.

Reviewer 2 Report

This review provides a clear overview about catalytic stereoselective dearomatization of pyridines. The Pineschi group has contributed important findings concerning catalysis with heterocycles and, generally, asymmetric catalysis.  Overall, the presented manuscript is sound and will be of interest to the readership of Molecules. Therefore, I recommend publication of the manuscript after fixing the following minor issues (mainly typos in the References section):

Page 20, line 684: "make" has to be substituted with "makes".

Page 23,lines 768-773: if possible, this period could be expressed more concisely.

-Page 24, lines 778-788: reference enumeration has to be corrected

-Page 23,  lines 740 and 750: References [74] and [75] are missing in the text.

-Page 24, line 783: the correct abbreviation for the journal has to be reported.

-Page 24, line 824: "(5)" should be deleted

 -Page 26, line 902: typo ")" at the end of the citation should be removed.

-Page 26, line 900: typo "]" should be removed.

-Page 26, line 910, 914, 922 and 932: ")" should be deleted.

Author Response

Rev 2:

This review provides a clear overview about catalytic stereoselective dearomatization of pyridines. The Pineschi group has contributed important findings concerning catalysis with heterocycles and, generally, asymmetric catalysis.  Overall, the presented manuscript is sound and will be of interest to the readership of Molecules. Therefore, I recommend publication of the manuscript after fixing the following minor issues (mainly typos in the References section):

Page 20, line 684: "make" has to be substituted with "makes". (done)

Page 23,lines 768-773: if possible, this period could be expressed more concisely. The final part of the conclusions has been rewritten

-Page 24, lines 778-788: reference enumeration has to be corrected     (done, we thank the Reviewer to realize this istakes)

-Page 23,  lines 740 and 750: References [74] and [75] are missing in the text. (fixed)

-Page 24, line 783: the correct abbreviation for the journal has to be reported. (done)

-Page 24, line 824: "(5)" should be deleted (done)

 -Page 26, line 902: typo ")" at the end of the citation should be removed. (done)

-Page 26, line 900: typo "]" should be removed. (done)

-Page 26, line 910, 914, 922 and 932: ")" should be deleted. (done)

Round 2

Reviewer 1 Report

Now it is okay to publish.